# Timing of Transfusion, not Hemoglobin Variability, Is Associated with 3-Month Outcomes in Acute Ischemic Stroke

**DOI:** 10.3390/jcm9051566

**Published:** 2020-05-21

**Authors:** Chulho Kim, Sang-Hwa Lee, Jae-Sung Lim, Mi Sun Oh, Kyung-Ho Yu, Yerim Kim, Ju-Hun Lee, Min Uk Jang, San Jung, Byung-Chul Lee

**Affiliations:** 1Department of Neurology, Chuncheon Sacred Heart Hospital, Chuncheon 24253, Korea; neurolsh@hallym.or.kr; 2Chuncheon Translational Research Center, Hallym University College of Medicine, Chuncheon 24252, Korea; 3Department of Neurology, Hallym University Sacred Heart Hospital, Anyang 14068, Korea; jaesunglim@hallym.or.kr (J.-S.L.); iyyar@hallym.or.kr (M.S.O.); ykh1030@hallym.or.kr (K.-H.Y.); 4Department of Neurology, Kangdong Sacred Heart Hospital, Seoul 05355, Korea; brainyrk@kdh.or.kr (Y.K.); leejuhun@kdh.or.kr (J.-H.L.); 5Department of Neurology, Dongtan Sacred Heart Hospital, Hwaseong 18450, Korea; mujang@hallym.or.kr; 6Department of Neurology, Kangnam Sacred Heart Hospital, Seoul 07440, Korea; neurojs@hallym.or.kr

**Keywords:** anemia, cerebral infarction, blood transfusion, red blood cells, outcome assessment

## Abstract

**Objectives:** This study aimed to investigate whether transfusions and hemoglobin variability affects the outcome of stroke after an acute ischemic stroke (AIS). Methods: We studied consecutive patients with AIS admitted in three tertiary hospitals who received red blood cell (RBC) transfusion (RBCT) during admission. Hemoglobin variability was assessed by minimum, maximum, range, median absolute deviation, and mean absolute change in hemoglobin level. Timing of RBCT was grouped into two categories: admission to 48 h (early) or more than 48 h (late) after hospitalization. Late RBCT was entered into multivariable logistic regression model. Poor outcome at three months was defined as a modified Rankin Scale score ≥3. Results: Of 2698 patients, 132 patients (4.9%) received a median of 400 mL (interquartile range: 400–840 mL) of packed RBCs. One-hundred-and-two patients (77.3%) had poor outcomes. The most common cause of RBCT was gastrointestinal bleeding (27.3%). The type of anemia was not associated with the timing of RBCT. Late RBCT was associated with poor outcome (odd ratio (OR), 3.55; 95% confidence interval (CI), 1.43–8.79; *p*-value = 0.006) in the univariable model. After adjusting for age, sex, Charlson comorbidity index, and stroke severity, late RBCT was a significant predictor (OR, 3.37; 95% CI, 1.14–9.99; *p*-value = 0.028) of poor outcome at three months. In the area under the receiver operating characteristics curve comparison, addition of hemoglobin variability indices did not improve the performance of the multivariable logistic model. Conclusion: Late RBCT, rather than hemoglobin variability indices, is a predictor for poor outcome in patients with AIS.

## 1. Introduction

Anemia is an independent predictor for mortality and cardiovascular disease in the general population [1]. The incidence of anemia in acute ischemic stroke (AIS) is 20–30%, and both extreme of admission hemoglobin has a U-shaped association with poor clinical outcomes [1,2]. Cerebral autoregulation enables the brain to maintain sufficient oxygenation in the blood when the cerebral perfusion pressure decreases [3]. However, this autoregulatory response to brain ischemia is already impaired in ischemic penumbra. Thus, anemia can have harmful effects on infarct growth or poor outcome [4,5].

As the erythropoietin trial has failed to validate the efficacy of outcomes in patients with AIS [6], red blood cell transfusion (RBCT) is the only way to normalize hemoglobin in patients with anemia. However, RBCT is associated with increased blood viscosity and a proinflammatory/prothrombotic state related with stored RBC and its additives [7,8]. The impact of hemoglobin status and RBCT on acute ischemic stroke is controversial [1]. In several studies, low hemoglobin status was associated with poor outcomes in patients with AIS; however, these studies focused on admission hemoglobin level and did not assess whether RBCT was performed during the admission [1]. There are several reports on the association between RBCT and AIS outcome. Moman et al. have reported that RBCT is associated with a longer hospital stay in patients with AIS with no difference in mortality [9]. They used propensity score matching to evaluate the impact of transfusion; however, they did not assess the hemoglobin status in all participants. Kellert et al. studied the association between RBCT and mortality and 3-month outcomes in patients with AIS admitted to a neurologic intensive care unit [10]. They reported that RBCT was not associated with mortality or 3-month outcomes. Further, they did not show variation in hemoglobin levels based on RBCT. In addition, one systematic review has suggested that anemia increases the mortality rate in patients with acute stroke; however, the association between RBCT and change in hemoglobin level were not evaluated [1]. Optimal hemoglobin management in acute stroke care should not only consider admission hemoglobin levels, but also the change in hemoglobin levels and RBCT during the hospitalization. Therefore, our aim is to assess the effect of type of anemia, timing of RBCT, and hemoglobin variability index during admission on the 3-month outcomes in patients with AIS, who received RBCT.

## 2. Material and Methods

### 2.1. Study Population

This retrospective observational study included prospectively collected stroke registry patients. Three tertiary teaching hospitals, part of the Clinical Research Center to Stroke—5 database and all laboratory data and clinical outcomes were prospectively collected, and central queries were revised bimonthly [11]. This study was approved by the Hallym University Hospital IRB (No. 2017-43), and an informed consent for registry enrollment and prospective outcome capture was given by all participants or next of kin. Our stroke registry included information of consecutive patients admitted within 7 days of the onset of stroke symptom. We screened patients diagnosed with AIS between January 2015 and December 2017. AIS was diagnosed if focal neurologic deficits persisted for more than 24 h and relevant lesions were confirmed by diffusion MRI. Patients without prospective outcome capture or relevant laboratory and clinical variables were excluded from this study.

### 2.2. Data Collection

The prospective registry data contained only admission hemoglobin level; therefore, all sequential hemoglobin levels during the hospital admission were extracted using the clinical data warehouse. The hemoglobin level was monitored according to the 2013 American Heart Association/American Stroke Association guideline. We used hemoglobin variability index as minimum, maximum, range (maximum-minimum), standard deviation (SD), coefficient of variance (CoV), median absolute deviation (MAD), and mean absolute change (MAC). Of these variability indices, MAC reflects a more temporal variation of the parameter than other variability indices [12]. Anemia was defined as a hemoglobin level of <13.0 g/dL for men and <12.0 g/dL for women according to World Health Organization criteria.

Whether the patient received RBCT was validated by filtering of the clinical data warehouse and retrospective chart review. We did not assess the administration of other blood products such as platelet concentrate or fresh frozen plasma. The criteria for determining the RBCT might vary from case to case, but they are commonly performed when hemoglobin falls below 8 g/dL. The timing of RBCT was divided into two categories—admission to 48 h (early) and >48 h after admission (late) [13]. The reason for RBCT was classified into five categories—gastrointestinal (GI) bleeding, cancer-related anemia, iron-deficiency anemia (IDA)/anemia of chronic disorder (ACD), surgery/procedure-related anemia, and others. GI bleeding was defined as the bleeding from the GI tract with an evidence of bleeding on endoscopy [14]. IDA was defined as an anemia with biochemical evidence of iron deficiency. ACD was defined as an anemia associated with chronic inflammatory, infectious disease, or malignancies [15]. Anemia associated with chronic kidney disease was also classified into this category. Cancer-related anemia was defined as anemia accompanied by a newly diagnosed, active, or metastatic cancer [16]. The determination of IDA/ACD or cancer-related anemia was mutually exclusive. For example, when the patient being treated with active cancer showed the IDA/ACD pattern, it was defined as cancer-related anemia. Surgery/procedure-related anemia was defined as newly developed anemia within 24 h after surgery or procedure without evidence of the other cause [17]. Finally, anemia without obvious causes was classified as other types of anemia (Figure 1).

### 2.3. Statistical Analysis

We compared the baseline characteristics of patients who received early and late RBC transfusion. The patients were divided into the good (mRS 0–2) and poor (mRS 3–6) group according to the 3-month outcome. Baseline demographic and clinical characteristics were compared using the χ2 or *t*-test (Mann–Whitney *U* test) as appropriate. Univariable logistic regression analysis was performed to assess the predictors for poor outcome. The multivariable logistic regression model was used for independent variables with a *p*-value of <0.05 in the univariable model or with the clinical relevance. We used four different multivariable models: model 1 adjusting for age and sex; model 2 adjusting for age, sex, and the Charlson comorbidity index (CCI); model 3 adjusting for age, sex, CCI, and NIHSS; and model 4 adjusting for age, sex, CCI, NIHHS, WBC count, and fasting blood glucose. For assessing the significance of hemoglobin variability parameters, model performance for each multivariable logistic regression analysis was performed using the area under the receiver operating characteristics curve (AUROC). Significant statistical differences among independent variables were considered with a *p*-value of 0.05 in multivariable models. All statistical analyses were performed using R (Foundation for Statistical Computing, Vienna, Austria, http://www.R-project.org).

## 3. Results

### 3.1. Baseline Characteristics

Among the 2698 patients with AIS, 592 (21.9%) were anemic at the time of admission and 132 (4.9%) received RBCT during the admission (Appendix A). Patients who received RBCT were older, more likely to be male, had history of previous stroke and smoking, and a higher stroke severity and cardioembolic cause of stroke than those who did not receive RBCT. The number of patients taking anticoagulants was higher in the RBCT group, but there was no difference in the previous use of antiplatelet agents before the index stroke between the two groups.

In total, 63 of 132 (47.7%) patients received early RBCT (Table 1). The mean age of patients who received RBCT during admission was the mean (± standard deviation) of 71.6 (±13.5) years, and 46.2% patients were men. Patients who received early RBCT were less likely to have previous strokes than those who received late RBCT. CCI and NIHSS score were not different between the early and late RBCT group. The proportion of patients with poor outcome (mRS >2) at 3 months was less in the early RBCT group than in the late RBCT group.

Patients who had poor outcomes had more severe stroke, shorter onset to admission time, and had a higher WBC count and fasting blood glucose level than those with good outcomes. Patients who received intravenous thrombolysis and RBCT were in the poor outcome group (Table 2).

### 3.2. Type of Anemia, RBC Transfusion and Hemoglobin Variability

GI bleeding (27.3%) was the most common cause of RBCT, followed by IDA/ACD (24.2%), surgery/procedure-related anemia (20.5%), and cancer-related anemia (15.2%). Most RBCT was performed within seven days of hospitalization (Figure 2a). The type of anemia was not associated with poor outcomes (*p* = 0.164 for chi-square, Table 2) and the timing of RBCT (Figure 2b and Table 3). However, patients with poor outcomes were found to have received RBCT later than those with good outcomes (*p* = 0.009 for chi-square, Table 2). The amount of RBCT showed left-shifted distribution and was higher in the good outcome group than in the poor outcome group, but it was not statistically significant (*p* = 0.337 for Wilcoxon signed-rank test).

During hospitalization, 2359 hemoglobin measurements were performed for the 132 patients receiving RBCT (Table 3). The median (interquartile range (IQR)) of hemoglobin measurements in each patient with RBCT was 13 (6–13) and the number of hemoglobin measurement was more frequent in the late RBCT group than in the early RBCT group (*p* <0.001). Among the hemoglobin variability parameters, IQR, range, and median absolute deviation (MAD) hemoglobin levels were lower and the mean absolute change (MAC) hemoglobin levels was higher in the early RBCT group than in the late RBCT group (Table 3). Mean, median, minimum, and maximum hemoglobin levels were higher in patients with poor outcomes than in those with good outcomes (Table 2). Other hemoglobin variability parameters including IQR, range, standard deviation (SD), MAD, coefficient of variation (CoV), and MAC hemoglobin were not different between the two groups.

### 3.3. Predictors for Poor Outcome

The proportion of patients with poor outcomes was 78.8% (104/132). In the univariable analysis, higher NIHSS score, late RBCT was associated with poor outcomes (odds ratio (OR), 3.55; 95% confidence interval (CI), 1.43–8.79) in univariable analysis. When we adjusted age, sex, CCI, NIHSS score, WBC count, and fasting blood sugar level, late RBCT was a significant predictor for poor outcomes (OR, 3.37; 95% CI, 1.14–9.99, Table 4).

Because of the high correlation between the hemoglobin variability parameters (Appendix A), the statistical significances of hemoglobin variability indices were compared to identify if the AUROC value indicating the model performance increased significantly when the hemoglobin variability parameters were added into the original logistic regression model. The performance of the original multivariable logistic regression was AUROC (0.883; 95% CI, 0.821–0.936, *p* <0.001). Figure 3 shows the model performance of each logistic regression model, which additionally included each hemoglobin variability parameter in the original model. However, there were no additional improvements in model performance when the hemoglobin variability indices were included in the original model.

## 4. Discussion

In this study, 21.9% patients with AIS were anemic at admission and 4.9% had received RBCT during hospitalization. Approximately 79% patients who received RBCT had a poor outcome at three months. The mean, median, minimum, and maximum hemoglobin levels were higher in patients with poor outcomes than in those with good outcomes. However, differences in hemoglobin variability indices, including IQR, range, SD, CoV, MAD, and MAC, did not differ between the two groups. Late RBCT was a significant predictor for poor outcome in patients with AIS in the multivariable model. However, hemoglobin variability indices were not associated with functional outcome in patients with AIS and RBCT.

We found that 5% AIS had received RBCT during hospitalization, and more than two-thirds of those had a poor outcome at three months. Additionally, late transfusion, rather than hemoglobin variability indices during hospitalization, was a significant predictor for poor outcome. Our study included all hemoglobin measurements performed during the hospital stay and differs from previous studies as we only investigated patients with AIS who had received RBCT. When analyzing the effect of hemoglobin on the AIS outcome, RBCT should be stratified RBCT or by analyzing only patients who had received RBCT.

Limited studies have assessed the relationship between the type of anemia and the functional outcome in patients with AIS. Ogata et al. have investigated the effect of GI bleeding in patients with AIS during hospitalization. Using the Fukuoka Stroke Registry, they showed that GI bleeding occurred most commonly within 1 week after the onset of stroke and was associated with poor outcome [21]. In our study, GI bleeding occurred in 38.8% patients, even after 7 days of hospitalization. As the antiplatelet agent regimen in AIS was changed and the characteristics of the patients varied in each study, there is a possibility that the prevalent period of GI bleeding may be different. However, RBCT performed to correct various causes of anemia not only during hospitalization but also at admission. In the retrospective observational study by Sharma et al., 28% of patients without anemia on admission developed anemia during admission [22]. In prospectively collected UK Regional Stroke Register data, hypochromic microcytic or normochromic normocytic anemia were associated with poor clinical outcomes in patients with AIS [23]. They concluded that the type of anemia is a salient indicator of comorbidity burden. Therefore, we suggest that the type of anemia or timing of RBCT could be an important predictor of functional outcome in patients with AIS.

In general, low or high hemoglobin levels adversely affect stroke outcomes [2,22,23]. The reason that our results did not show the U-shaped relationship between hemoglobin levels and stroke outcomes was due to the difference in patient population. The previous reports have assessed admission hemoglobin levels for all patients admitted with AIS, and this study only included patients with AIS who received RBCT during the hospital stay. Likewise, when analyzing only patients who had received RBCT, we can hypothesize that the other variables, including the type of anemia, had more impact on stroke outcome than the initial hemoglobin level.

RBCT had a poorer outcome than those with early RBCT due to admission hemoglobin level not differing between early and late RBCT group; however, the IQR and range hemoglobin were higher in late transfusion group than in early RBCT group in our report. Furthermore, the number of hemoglobin measurements performed during the hospital stay was an average of 7 times in early RBCT group, but an average of 19 measurements in late RBCT group. Based on these observations, the change in hemoglobin status was higher and more abrupt in the late transfusion group than in the early RBCT group, and it can be expected that more frequent hemoglobin measurements in late RBCT group were made to monitor this rapid change. The autoregulatory mechanism for maintaining cerebral blood flow is already lost in infarct core, and that mechanism is already maximized in the penumbral area [24]. Therefore, the rapid drop in hemoglobin may further exacerbate the oligemia and cause infarct growth in penumbral area. In a study by Bellwald et al., decreased hemoglobin level after hospital arrival was associated with the amount and velocity of infarct growth in patients with AIS [25]. We did not assess infarct growth of our participants; rapid alteration of hemoglobin status in the late RBCT group may have a worse effect on the cerebral autoregulatory mechanism, which can exacerbate stroke outcome. Second, although the type of anemia was not statistically different between early and late RBCT group (*p* = 0.080) in our data, different cause of RBCT may affect stroke outcome. Surgery/procedure-related anemia was higher in the early RBCT group, and GI bleeding was higher in the late RBCT group in our study. However, our study did not include a large number of patients who received RBCT (only 5% in total AIS population), it should be reassessed in a larger prospective study whether this type of anemia affected stroke outcome.

In general population, history of RBCT is associated with 1.6-fold increase in the risk of ischemic stroke [26]. This association is explained by the fact that stored RBC increases blood viscosity, and decreases nitric oxide concentration, vasoconstriction, and platelet activation [1]. On the other hand, low hemoglobin is inversely correlated with initial infarct volume or infarct growth, and the author suggested that RBCT would be beneficial for recovery of stroke [1]. However, restrictive transfusion strategy in patients of cardiovascular diseases was not inferior compared to liberal strategy in two systematic review [1]. In our data, transfusion amount did change between the good and poor outcome group, though those with good outcome had low mean, median, minimum, and maximum hemoglobin compared to those with poor outcome. Our study did not directly assess the exact transfusion strategy because the study design was retrospective in nature. However, we suggested that restrictive transfusion strategies can reduce the thrombotic complication and maximize the beneficial effect by RBCT compared to the liberal strategy in AIS patients.

In our study, hemoglobin variability indices did not affect the functional outcome in patients with AIS. In the previous report, minimum or maximum hemoglobin level was associated with worse outcome [1]. However, these hemoglobin parameters did not affect the stroke outcome in our study. Kellert et al. studied the impact of low hemoglobin level and transfusion in neurologic intensive care unit patients and found that hemoglobin parameters were not associated with in-hospital mortality or 90-day functional outcomes, but they were associated with length of intensive care unit length of stay and duration of mechanical ventilation [10]. The author suggested that the impact of hemoglobin parameters in neurologically severe patients might be reduced by the important predictors such as stroke severity. In our study, patients with RBCT had more severe stroke than those without RBCT (median NIHSS score 13 vs. 3). Our study also suggests that stroke severity, rather than hemoglobin parameters, is an important predictor for poor outcome in patients with severe ischemic stroke. However, as our study and Kellert’s study had a small sample size, larger prospective studies are needed to confirm these associations.

Our study had some limitations. First, our study was a small sampled-sized retrospective observation, and therefore there is a chance of selection bias and residual confounding. However, the incidence of anemia and the proportion who had received RBCT during hospitalization were comparable to other studies on patients with AIS [1]. Second, the effect of RBCT on functional outcome was likely to be underestimated because we only collected RBCT data, which were performed only during hospitalization. However, anemia usually developed 2–11 days following admission in patients with AIS [27]. Therefore, only several patients would receive RBCT after discharge.

Despite these limitations, our study had several strengths. First, we minimized residual confounding by including information such as stroke severity, type of anemia, and timing and amount of RBCT. Second, the characteristics of patients with AIS who received RBCT differ significantly compared to those who did not receive RBCT. If the rare event (such as patients with RBCT; ~5% of all AIS patients) is evaluated with logistic regression method, the results may be vulnerable to biases [28]. We solved this problem by analyzing the binary outcome only in patients with RBCT and minimized the interaction between anemia and RBCT transfusion. Third, cerebral perfusion can be changed dynamically depending on the degree of anemia and whether the RBCT is performed or not. We evaluated all hemoglobin measurements during the hospitalization. In addition, we analyzed the overall hemoglobin parameters such as SD, CoV, and MAD, and temporal variation parameter such as MAC. In addition, we identified all bleeding events during hospitalization and reflected them in the type of anemia variable.

## 5. Conclusions

Late RBCT was associated with 3-month poor outcome in patients with AIS. To verify this, a larger prospective study is needed for assessing the type of anemia and cause of RBCT, and the fluctuation of hemoglobin status during the admission.

## Figures and Tables

**Figure 1 jcm-09-01566-f001:**
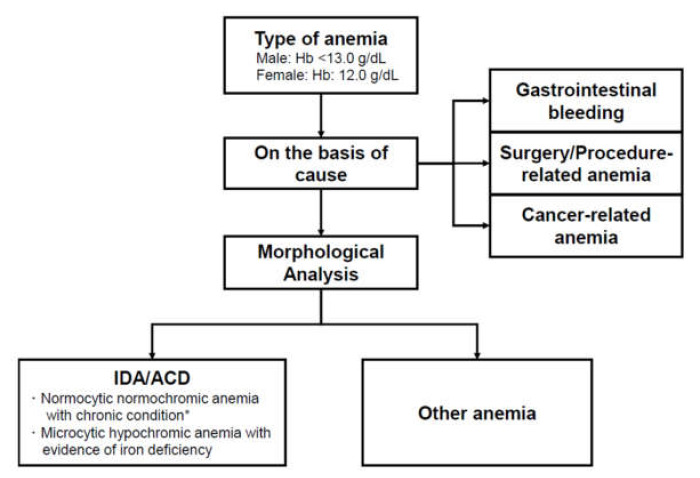
Type of anemia according to cause of anemia or morphological analysis of erythrocytes. * Patients with anemia of chronic disease were classified as cancer-related anemia when they had active cancer. Hb: hemoglobin; IDA: iron-deficiency anemia; ACD: anemia of chronic disease. We included the additional laboratory results that can affect the hemoglobin levels and anemia status: white blood cell (WBC) and platelet counts; blood urea nitrogen, creatinine, and blood glucose levels; international normalized ratio; and blood pressure. The functional outcome was assessed by modified Rankin Scale (mRS) score at 3 months [18], and stroke severity was measured using the National Institute of Health Stroke Scale (NIHSS) score at admission [19]. The primary outcome was poor outcome at 3 months, which was defined the mRS score of 3–6 [20]. Secondary analysis was performed to assess the significance of each hemoglobin variability parameters during admission in poor outcome prediction.

**Figure 2 jcm-09-01566-f002:**
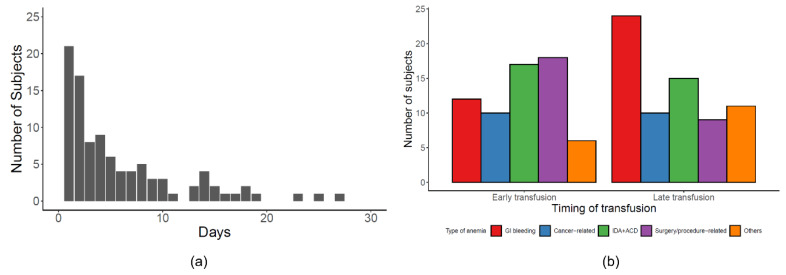
The timing of red blood cell transfusion and the relationship of type of anemia between early and late red blood cell transfusion: (**a**) The frequency of red blood cell transfusion performed after hospitalization (X-axis means the timing (days) of RBCT after admission; Y-axis means the number of subjects who received red blood cell transfusion). (**b**) The differences of proportion in type of anemia according to the timing of the red blood cell transfusion (early vs. late).

**Figure 3 jcm-09-01566-f003:**
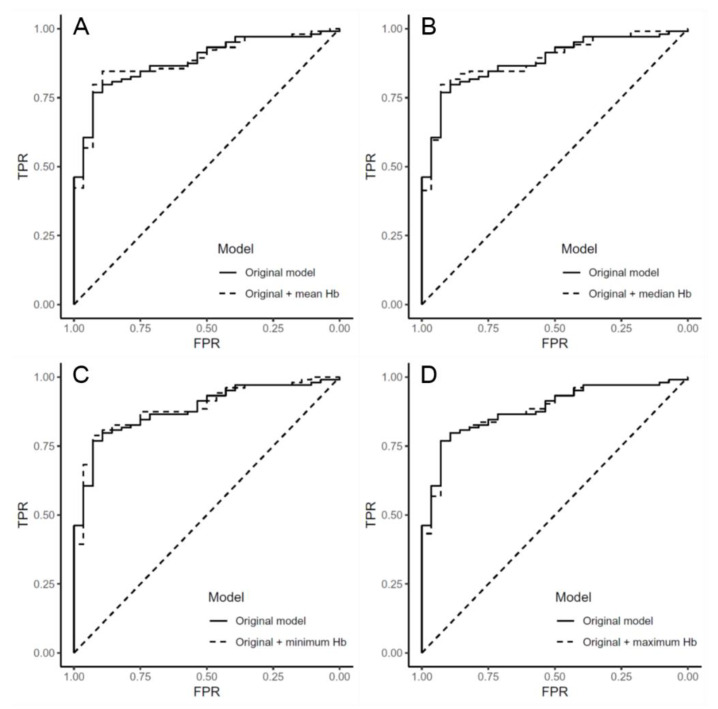
Receiver operating characteristics curve showing performances of original logistic regression model and the other models with hemoglobin variability parameter. The original model was the final logistic regression model in Table 4. The area under the curve of the Receiver Operating Characteristic (AUROC) of the original model was 0.883. AUROC of mean (**A**), median (**B**), minimum (**C**), and maximum (**D**) hemoglobin-adjusted models were 0.884, 0.885, 0.889, and 0.882, respectively. TPR: true positive rate; FPR: false positive rate; Hb: hemoglobin.

**Table 1 jcm-09-01566-t001:** Baseline characteristics of the participants.

Parameters	Early Transfusion (*n* = 63)	Late Transfusion (*n* = 69)	*p*
Age, years	68.6 ± 16.3	74.4.1 ± 9.7	0.450
Male	30 (47.6%)	31 (44.9%)	0.893
Past medical history			
Stroke	16 (25.4%)	30 (43.5%)	0.002
Hypertension	41 (65.1%)	52 (75.4%)	0.270
Diabetes	17 (27.0%)	23 (33.3%)	0.546
Hyperlipidemia	18 (28.6%)	20 (29.0%)	0.985
Current smoking	19 (30.2%)	12 (17.4%)	0.128
Charlson comorbidity index	5.0 (3.0–7.0)	5.0 (4.0–7.0)	0.221
Stroke subtype			0.116
Cardioembolic	14 (22.2%)	25 (36.2%)	
Non-cardioembolic	49 (77.8%)	44 (63.8%)	
NIHSS score	9.0 (2.5–16.0)	13.0 (6.0–18.0)	0.091
Thrombolysis	9 (14.3%)	6 (8.7%)	0.461
Onset to visit time, hour	3.7 (1.2–30.0)	4.9 (1.0–30.5)	0.879
Laboratory parameter			
WBC, 10^3^/μL	9.1 ± 4.5	9.4 ± 4.3	0.660
Platelet, 10^3^/μL	280 ± 152	234 ± 111	0.050
BUN, mg/dL	22.2 ± 15.6	22.5 ± 16.9	0.926
Creatinine, mg/dL	1.2 ± 1.3	1.3 ± 1.2	0.614
Total cholesterol, mg/dL	147.0 ± 49.7	160.0 ± 48.3	0.130
TG, mg/dL	95.5 ± 55.2	113.0 ± 55.5	0.078
HDL, mg/dL	45.6 ± 13.2	42.0 ± 11.8	0.108
LDL, mg/dL	90.3 ± 41.2	92.6 ± 42.9	0.759
FBS, mg/dL	133.0 ± 57.3	139.0 ± 55.9	0.574
INR	1.2 ± 0.7	1.3 ± 0.9	0.816
Systolic BP, mmHg	140 ± 25	140 ± 28	0.997
Diastolic BP, mmHg	81.8 ± 13.7	78.9 ± 17.3	0.284
History of antithrombotics usage	23 (36.5%)	34 (49.3%)	0.193
Poor outcome (mRS >2)	43 (68.3%)	61 (88.4%)	0.009

Categorical variables are represented by the number (column percent), and continuous variable are represented by mean (± standard deviation) or median (interquartile range) as appropriate. SD: standard deviation; iqr: interquartile range; NIHSS: National Institute of Health Stroke Scale; WBC: white blood cell, BUN: blood urea nitrogen; TG: triglycerides; HDL: high-density lipoprotein; LDL: low-density lipoprotein; FBS: fasting blood sugar; INR: international normalized ratio; BP: blood pressure; mRS: modified Rankin Scale.

**Table 2 jcm-09-01566-t002:** The comparison of clinical and laboratory parameters between good and poor outcome group.

Parameters	Poor (*n* = 104)	Good (*n* = 28)	*p*
Age, years	70.1 ± 14.9	72.0.1 ± 13.2	0.514
Male	43 (41.3%)	18 (64.3%)	0.051
Past medical history			
Stroke	39 (37.5%)	7 (25.0%)	0.313
Hypertension	75 (72.1%)	18 (64.3%)	0.567
Diabetes	34 (32.7%)	6 (21.4%)	0.358
Hyperlipidemia	29 (27.9%)	9 (32.1%)	0.836
Current smoking	21 (20.2)	10 (35.7)	0.142
Stroke subtype			0.718
Cardioembolic	32 (30.8%)	7 (25.0%)	
Non-cardioembolic	70 (68.6%)	23 (76.7%)	
NIHSS, score	13.0 (7.0–18.0)	3.0 (1.0–4.5)	<0.001
Thrombolysis	15 (14.4%)	0 (0.0%)	0.072
onset to visit time, hour	3.2 (0.9–26.0)	12.9 (3.2–61.7)	0.012
Laboratory parameter			
WBC, 10^3^/μL	9.7 ± 4.6	7.4 ± 2.7	0.011
Platelet, 10^3^/μL	246 ± 127	292 ± 152	0.102
BUN, mg/dL	21.5 ± 14.8	25.8 ± 20.6	0.211
Creatinine, mg/dL	1.2 ± 1.3	1.3 ± 0.9	0.878
Total cholesterol, mg/dL	157.0 ± 50.2	142.0 ± 44.3	0.178
TG, mg/dL	103.0 ± 53.5	109.0 ± 63.4	0.558
HDL, mg/dL	44.5 ± 12.1	40.3 ± 13.7	0.120
LDL, mg/dL	93.6 ± 43.5	84.0 ± 35.5	0.290
FBS, mg/dL	143.0 ± 58.7	110.0 ± 37.5	0.006
INR	1.3 ± 0.9	1.1 ± 0.1	0.274
Systolic BP, mmHg	143.0 ± 26.6	132.0 ± 24.4	0.058
Diastolic BP, mmHg	81.2 ± 16.2	77.0 ± 13.4	0.211
History of antithrombotics usage	46 (44.2%)	11 (39.3%)	0.780
Number of Hb measure	13.0 (6.0–23.5)	8.0 (5.8–19.3)	0.309
Admission Hb, g/dL	9.7 ± 2.6	8.8 ± 2.5	0.129
Hb variability parameter			
Mean, g/dL	10.2 ± 1.5	9.4 ± 1.2	0.012
Median, g/dL	10.2 ± 1.5	9.5 ± 1.2	0.018
Minimum, g/dL	8.0 ± 1.8	7.2 ± 1.4	0.025
Maximum, g/dL	12.2 ± 1.9	11.3 ± 1.7	0.026
IQR, g/dL	1.6 ± 0.9	1.6 ± 0.9	0.806
Range, g/dL	4.2 ± 1.9	4.1 ± 1.6	0.889
SD, g/dL	1.3 ± 0.5	1.4 ± 0.5	0.640
MAD, g/dL	1.1 ± 0.7	1.2 ± 0.7	0.529
CoV, %	12.9 ± 4.8	14.4 ± 4.5	0.134
MAC, g/dL	0.7 ± 0.4	0.8 ± 0.2	0.165
Type of anemia			0.164
GI bleeding	24 (23.1)	12 (42.9)	
Cancer-related	18 (17.3)	2 (7.1)	
IDA or ACD	24 (23.1)	8 (28.6)	
Surgery/Procedure-related	23 (22.1)	4 (14.3)	
Others	15 (14.4)	2 (7.1)	
Transfusion amount, mL	400 (400–800)	800 (400–1140)	0.337
Timing of transfusion			0.009
Early (≤ 48 h)	43 (41.3%)	20 (71.4%)	
Late (> 48 h)	61 (58.7%)	8 (28.6%)	

Categorical variables are represented by the number (column percent) and continuous variable are represented by mean (± standard deviation) or median (interquartile range) as appropriate. NIHSS: National Institute of Health Stroke Scale; WBC: white blood cell; BUN: blood urea nitrogen; TG: triglycerides; HDL: high-density lipoprotein; LDL: low-density lipoprotein; FBS: fasting blood sugar; INR: international normalized ratio; BP: blood pressure; Hb: hemoglobin; IQR: interquartile range; SD: standard deviation; MAD: median absolute deviation; CoV: coefficient of variation; MAC: mean absolute change; GI: gastrointestinal; IDA: iron deficiency anemia; ACD: anemia of chronic disorder.

**Table 3 jcm-09-01566-t003:** The comparison of hemoglobin variability parameters and type of anemia between early and late transfusion group.

Parameters	Early Transfusion (*n* = 63)	Late Transfusion (*n* = 69)	Total (*n* =132)	*p*
Number of Hb measure, number	7.0 (4.0–14.0)	19.0 (10.0–30.0)	13.0 (6.0–23.0)	<0.001
Admission Hb, mg/dL	9.4 ± 2.7	9.6±2.5	9.5 ± 2.6	0.639
Hb variability parameter				
Mean, mg/dL	10.1 ± 1.8	9.9 ± 1.1	10.0 ± 1.5	0.391
Median, mg/dL	10.3 ± 1.9	9.9 ± 1.1	10.1 ± 1.5	0.109
Minimum, mg/dL	8.2 ± 2.1	7.6 ± 1.3	7.9 ± 1.7	0.065
Maximum, mg/dL	11.7 ± 1.9	12.3 ± 1.8	12.0 ± 1.9	0.052
IQR, mg/dL	1.3 ± 0.8	1.8 ± 1.0	1.9 ± 0.6	0.004
Range, mg/dL	3.5 ± 1.6	4.7 ± 1.8	4.1 ± 1.8	<0.001
SD, mg/dL	1.3 ± 0.5	1.4 ± 0.5	1.3 ± 0.5	0.292
MAD, mg/dL	1.0 ± 0.7	1.3 ± 0.7	0.5 ± 0.5	0.025
CoV, %	12.6 ± 4.8	13.7 ± 4.7	13.2 ± 4.8	0.212
MAC, mg/dL	0.8 ± 0.4	0.6 ± 0.3	0.4 ± 0.3	0.004
Type of anemia				0.080
GI bleeding	12 (19.0)	24 (34.8)	36 (27.3)	
Cancer-related	10 (15.9)	10 (14.5)	20 (15.2)	
IDA or ACD	17 (27.0)	15 (21.7)	32 (24.2)	
Surgery/Procedure-related	18 (28.6)	9 (13.0)	27 (20.5)	
Others	6 (9.5)	11 (16.0)	17 (12.9)	
Transfusion amount, mL	400 (400–800)	640 (400–1120)	400 (400–840)	0.448

Categorical variables are represented by the number (column percent) and continuous variable are represented by mean (± standard deviation) or median (interquartile range) as appropriate. Hb: hemoglobin; IQR: interquartile range; SD: standard deviation; MAD: median absolute deviation; CoV: coefficient of variation; MAC: mean absolute change; GI: gastrointestinal; IDA: iron deficiency anemia; ACD: anemia of chronic disorder.

**Table 4 jcm-09-01566-t004:** The predictors of poor outcome in multivariable logistic regression analysis according to the timing of the transfusion.

	Crude Model		Model 1		Model 2		Model 3		Model 4	
	OR (95% CI)	*p*	OR (95% CI)	*p*	OR (95% CI)	*p*	OR (95% CI)	*p*	OR (95% CI)	*p*
Late transfusion	3.55 (1.43–8.79)	0.006	3.61 (1.42–9.19)	0.007	3.65 (1.43–9.29)	0.007	3.21 (1.14–9.09)	0.028	3.37 (1.14–9.99)	0.028
Age			1.00 (0.97–1.03)	0.983	1.00 (0.96–1.03)	0.806	0.99 (0.95–1.04)	0.694	0.98 (0.93–1.03)	0.403
Male			0.38 (0.16–0.93)	0.035	0.38 (0.15–0.92)	0.033	0.43 (0.16–1.17)	0.099	0.42 (0.15–1.21)	0.109
CCI					1.04 (0.86–1.27)	0.677	1.06 (0.84–1.35)	0.629	1.07 (0.83–1.39)	0.592
NIHSS							1.22 (1.11–1.34)	<0.001	1.19 (1.08–1.31)	<0.001
WBC, 10^3^/μL									1.16 (0.98–1.37)	0.089
FBS, mg/dL									1.01 (1.00–1.02)	0.189

Model 1 adjusted for age and sex; Model 2 included variables in Model 1 plus Charlson comorbidity index; Model 3 included variables in Model 2 plus National Institute of Health Stroke Scale score; Model 4 included variables in Model 3 plus white blood cell count and fasting blood sugar. OR: odds ratio; CI: confidence interval; CCI: Charlson comorbidity index; NIHSS: National Institute of Health Stroke Scale; WBC: white blood cell; FBS: fasting blood sugar.

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
