# Peer review of "Timing of Transfusion, not Hemoglobin Variability, Is Associated with 3-Month Outcomes in Acute Ischemic Stroke"

_jcm, 2020, doi:10.3390/jcm9051566_

Round 1
Reviewer 1 Report
In this manuscript, authors have investigated whether transfusion and hemoglobin variability affect the outcome of stroke after acute ischemic stroke. It is a very interesting study to see how HB level and timing of transfusion affect AIS outcomes.
- Abstract:
- The hemoglobin variability is not mentioned in the results section
- Introduction:
- Throughout the text, authors should review the use of abbreviations, especially RBC, RBCT or MRI
- Use red blood cell transfusion (RBTC) iinstead of red blood cell (RBC) transfussion (RBCT)
- The aim is not clearly defined. It must be rewritten
- Material and Methods
- It is not clear what the type of study is: prospective or rerospective
- Which is the reason to exclude patients without relevant laboratory and clinical variables?
- Bibliographic references of Rankin Scale or NIHSS are missing How was the follow-up of the patients along three months? The choice of 3-6 values on the Rankin scale as a poor outcome, is it based on any study? The statical analysis must be rearranged Is the multivariate analysis adjusted for any variable? Are different models created? Indicate them Specify what is the p-value considered in the data analysis
- Results
- The result mus be rearranged to facilitate their compression
- In the line 190, the authors say "surgery/procedure-related anemia (0,5%)..." Is this value correct? The types of anemia are ordered from the most frequent to the least frequent
- In the table 3 the timing pf RBCT is no described
- The data in table 3 should be merged into table 1
- Clarify the figure 2 legend
- In the lines 213, the authors say "... and number of hemoglobin measurement was more frequent in the poor outcome group than in good outcome group (p <0.001)". Based on table 2 results, that p-value should be 0.309, not being significant
- In the line 224, the authors say: "The proportion of patients with poor outcomes was 77.3% (102/132)" Is these data correct?
- In the line 225, the authors repeat repeat what they have said on line 175. Delete it
- In Table 4, the authors should include the rest of the variables included in the different models with their odds ratio value.
- Conclussion
- It is not in line with what is stated in the rest of the study. Redone it
Author Response
Thanks for the comment. We do our best to correct our manuscript as your suggestion. Especially, there were some big mistakes. Thanks for pointing this out.
In this manuscript, authors have investigated whether transfusion and hemoglobin variability affect the outcome of stroke after acute ischemic stroke. It is a very interesting study to see how HB level and timing of transfusion affect AIS outcomes.
1.Abstract: The hemoglobin variability is not mentioned in the results section
Thanks for the comment. In our analysis of binary logistic regression including Hb variability indices (Figure 3), those variability were not statistically significant. Therefore, we summarized these results as below.
In the area under the receiver operating characteristics curve comparison, addition of hemoglobin variability indices did not improve the performance of the multivariable logistic model.
2.Introduction: Throughout the text, authors should review the use of abbreviations, especially RBC, RBCT or MRI
Use red blood cell transfusion (RBCT) instead of red blood cell (RBC) transfusion (RBCT)
: We corrected the expression as the reviewer’s recommendation.
The aim is not clearly defined. It must be rewritten.
As the reviewer’s suggestion, we amended the aim of our study as below.
Therefore, our aim is to assess the effect of type of anemia, timing of RBCT, and hemoglobin variability index during admission on the 3-month outcomes in patients with AIS, who received RBCT.
3.Material and Methods It is not clear what the type of study is: prospective or retrospective.
This is a retrospective observational study. We used prospectively collected registry data, but the research hypothesis was written in the middle of patient recruitment, not in the beginning of the patient enrollment.
Which is the reason to exclude patients without relevant laboratory and clinical variables?
We excluded 27 patients, most of whom had not visited outpatient clinic after discharge or were not interviewed by telephone. We considered multiple imputation method to adjust the missing variables. However, mRS was an important outcome variable in our study. Therefore, we did not use this missing-adjusting method, and excluded 27 patients.
Bibliographic references of Rankin Scale or NIHSS are missing.
Thanks for the comment. We added references of the mRS and the NIHSS.
How was the follow-up of the patients along three months?
Most of the patients were usually hospitalized for 10 days. After discharge, they were followed up in the outpatient clinic.
The choice of 3-6 values on the Rankin scale as a poor outcome, is it based on any study?
In most studies analyzing stroke outcomes, mRS ≥ 3 is defined as poor outcome. We added a reference to this.
The statistical analysis must be rearranged. Is the multivariate analysis adjusted for any variables? Are different models created? Indicate them Specify what is the p-value considered in the data analysis
As the reviewer’s suggestion, we amended the “Statistical Analysis” section as below.
We compared the baseline characteristics of patients who received early and late RBC transfusion. The patients were divided into the good (mRS 0-2) and poor (mRS 3-6) group according to the 3-month outcome. Baseline demographic and clinical characteristics were compared using the χ2 or t-test (Mann-Whitney U test) as appropriate. Univariable logistic regression analysis was performed to assess the predictors for poor outcome. The multivariable logistic regression model was used for independent variables with a p value of <0.05 in the univariable model or with the clinical relevance. We used 4 different multivariable models: model 1 adjusting for age and sex, model 2 adjusting age, sex, and the Charlson comorbidity index (CCI), model 3 adjusting for age, sex, CCI, and NIHSS, model 4 adjusting, age, sex, CCI, NIHHS and WBC count and fasting blood glucose. For assessing the significance of hemoglobin variability parameters, model performance for each multivariable logistic regression analysis was performed using the area under the receiver operating characteristics curve (AUROC). Significant statistical differences among independent variables were consider with a p-value of 0.05 in multivariable models. All statistical analyses were performed using R (R Development Core Team 2015).
4.Results The result must be rearranged to facilitate their compression
In the line 190, the authors say "surgery/procedure-related anemia (0,5%)..." Is this value correct? The types of anemia are ordered from the most frequent to the least frequent.
It was a critical mistake. We corrected the exact proportion of surgery/procedure-related anemia from 0.5% to 20.5%.
In the table 3 the timing of RBCT is no described
Table 3 shows the differences of independent variables between early vs late RBCT groups. The description of the result is represented in “Result 3.2” section.
The data in table 3 should be merged into table 1
Thanks for the comment. When Table 3 is attached to Table 1, the variables were too long to depict in one page. And, we thought that it is easy for readers to read easily the parameters related to hemoglobin, so the tables are not combined.
Clarify the figure 2 legend
We clarify the legend of the Figure 2. You can see a clear legend in the original figure file.
In the lines 213, the authors say "... and number of hemoglobin measurement was more frequent in the poor outcome group than in good outcome group (p <0.001)". Based on table 2 results, that p-value should be 0.309, not being significant.
Since this section describes the early and late RBCT separately, we have changed this sentence as follows.
The median (interquartile range [IQR]) of hemoglobin measurements in each patient with RBCT was 13 (6-13) and the number of hemoglobin measurement was more frequent in the late RBCT group than in the early RBCT group (p < 0.001).
In the line 224, the authors say: "The proportion of patients with poor outcomes was 77.3% (102/132)" Is these data correct?
: We corrected this critical mistake as below.
The proportion of patients with poor outcomes was 78.8% (104/132).
In the line 225, the authors repeat what they have said on line 175. Delete it
: Thanks for the comment. We deleted the redundant expressions in the line 225
In Table 4, the authors should include the rest of the variables included in the different models with their odds ratio value.
: Thanks for the comment. We corrected table 4 as the reviewer’s suggestion.
5.Conclussion It is not in line with what is stated in the rest of the study. Redone it
: As the reviewer’s suggestion, we clarify our conclusion and future perspectives.
Late RBCT was associated with the 3-month poor outcome in patients with AIS. To verify this, more large prospective study is needed for assessing the type of anemia and cause of RBCT, and the fluctuation of hemoglobin status during the admission.

Reviewer 2 Report
Dear Authors-
It looks good and all concerns are addressed except TABLE 1, 2, 3, and 4 “p-value” in 3rd column. Please make headings.
Thanks
Author Response
It looks good and all concerns are addressed except TABLE 1, 2, 3, and 4 “p-value” in 3rd column. Please make headings.
: Thanks for the comment. We corrected this as the reviewer’s suggestion.
This manuscript is a resubmission of an earlier submission. The following is a list of the peer review reports and author responses from that submission.
Round 1
Reviewer 1 Report
Dear Authors and Editors-
In this manuscript, authors have investigated whether transfusion and hemoglobin variability affect the outcome of stroke after acute ischemic stroke. It is a very interesting study to see how HB level and timing of transfusion affect AIS outcomes.
My suggestions are below-
Abstract: Missing structure-Intro/Objective, Methods, Results, Conclusion-Statistical Analysis
Methods: t-test type pair? how median significance is determined Wilcoxon test? please mention.
Methods: Please add a subsection on Outcomes defining primary and secondary outcomes and their definitions
Results: “As a result of high correlation between the hemoglobin variability parameters (Figure S1), the statistical significances of hemoglobin variability indices were compared to identify if the AUROC value indicating the model performance increased significantly when the hemoglobin variability parameters were added into the original logistic regression model. The performance of the original multivariable logistic regression was AUROC (0.883; 95% CI, 0.821-0.936, p <0.001). Figure S2 shows the model performance of each logistic regression model, which additionally included each hemoglobin variability parameter into the original model. However, there were no additional improvement of model performance when the hemoglobin variability indices were included in the original model. “
As the title of the manuscript includes Hemoglobin variability role on the outcome that is why the regression analysis model of it should be in the original file and not in the supplemental file.
Results: Tables in bracket mentioned numbers are fq%, SE? Mean/Median+ SE/SD please mention the details.. for e.g. for age variables it is mean/median + SD/SE; for stroke variable it is column % or fq%
Table 2: Poor vs Good outcome table Hb variability and Early transfusion chi (categorical)/t(mean)/wilcoxon(median) are there but late transfusion details are missing. Also, Early transfusion % sign in a good outcome is missing.
Please use column % if you have not used as fq% can not determine exact relationship comparison between 2-groups (also mention that you used column% to compare in chi-square tables)
Thanks
Reviewer 2 Report
Título: El título no es representativo del estudio. Revisalo
Introducción: debe proporcionar más información sobre la situación actual del tema
Método: explique el procedimiento para reclutar pacientes y evaluar los resultados a los 3 meses
Discusión: No puede hacer algunas afirmaciones, teniendo en cuenta el tipo de estudio que ha realizado. Revisalo
Revisa el texto, hay párrafos sin referencias
Verifique las abreviaturas, no se explica todo su significado
Reviewer 3 Report
In this paper, Chulho Kim et al. demonstrated that timing of transfusion, not hemoglobin variability, is related to 3 month poor outcome. The project in general is interesting and raises an important problem.
However, there are a lot of issues to discuss.
In line 24 -25: "Timing of RBCT was divided into 2 categories: admission to 48 24 hours (early) or more than 48 hours (late) after hospitalization"
In line 92: "The timing of RBCT 91 was divided into 2 categories: admission to 48 hours and more than 48 hours after the admission"
Which one is correct??
Line 27: received 400 (400-840) ml - 400 is a mean? median? average? it is a mistake here.
Line 30-32 should be p-values added.
Lines 93-105- division of anemies is contractual and very complicated and this part is very difficult to understand.I suggest another division that could be more understood and transparent.
Lines 109-110 when two scores were assesed? during admission, discharge?
The lack of this information is crucial because the clinical condition may change during treatment.
In Table 2 significantly higher levels of hemoglobin are related to good clinical outcome- and in the conclusion in lines 243-245 "In our data, hemoglobin variability indices did not affect the functional outcome of AIS patients. In the previous report, minimum or maximum hemoglobin level were associated with worse outcom . However, these hemoglobin parameters did not affect stroke outcome in our study"
- the data is inconsistent. In general the reported data is in contradictory to the previous results that showed that lower level of hemoglobin are related to poor clinical outcome.
How the authors can explain these differences?
In Table 2 we have both information about hemoglobin- mean and median the first one belongs to parametric tests, the second to non-parametric.
Why the both information are together?
Figure 1 should be in color and in better resolution - at this moment the legends are not visible and clear.
Lines 209-216- the authors repeat results.
In first part of discussion it should be underlining the novelties and discoveries demonstrated in this study!!!
Line 231- the only citation is about subarachnoidal hemorrhage - that is not a subject of this study and should not be correlated with ischemic stroke.
Line 81-82 "Hemoglobin was not routinely measured during the admission but was only measured when needed 81 for general care including monitoring of bleeding, transfusion or infection"- it is impossible, Hb level is one the most important parameters in blood so it should be always evaluated in stroke subjects during admission!!!
Major comments:
The most important conclusion from this paper is relationship of late transfusion and poor clinical outcome.
In conclusion authors did not focus at this point- what is the idea for this relationship? what are clinical conseqences? how this relationship can be explained? how timing of transfusion can affects clinical outcome?
The authors did not emphasize the clinical and practical meaning of their major finding.
I was wondering what is the idea of division of transfusion into early and late?? Why some patients had early and some had late transfusion? Was it depend on the time of the serious bleeding? Was it related to clinical emergency or deterioration?
I suppose that above problems could be associated with retrospective type of this research based on the registry. No data may be available.
What about the informed consent? was it written retrospectively?
Overall, there are too many serious problems and too many mistakes and misunderstatements.